# Cdc42 Couples T Cell Receptor Endocytosis to GRAF1-Mediated Tubular Invaginations of the Plasma Membrane

**DOI:** 10.3390/cells8111388

**Published:** 2019-11-04

**Authors:** Pascal Rossatti, Luca Ziegler, Richard Schregle, Verena M. Betzler, Manuela Ecker, Jérémie Rossy

**Affiliations:** 1Biotechnology Institute Thurgau (BITg) at the University of Konstanz, 8280 Kreuzlingen, Switzerland; 2Department of Biology, University of Konstanz, 78457 Konstanz, Germany; 3EMBL Australia Node in Single Molecule Science, School of Medical Sciences, University of New South Wales, Sydney 2052, Australia

**Keywords:** T cell receptor, clathrin-independent endocytosis, CLIC, Cdc42, GRAF1

## Abstract

T cell activation is immediately followed by internalization of the T cell receptor (TCR). TCR endocytosis is required for T cell activation, but the mechanisms supporting removal of TCR from the cell surface remain incompletely understood. Here we report that TCR endocytosis is linked to the clathrin-independent carrier (CLIC) and GPI-enriched endocytic compartments (GEEC) endocytic pathway. We show that unlike the canonical clathrin cargo transferrin or the adaptor protein Lat, internalized TCR accumulates in tubules shaped by the small GTPase Cdc42 and the Bin/amphiphysin/Rvs (BAR) domain containing protein GRAF1 in T cells. Preventing GRAF1-positive tubules to mature into endocytic vesicles by expressing a constitutively active Cdc42 impairs the endocytosis of TCR, while having no consequence on the uptake of transferrin. Together, our data reveal a link between TCR internalization and the CLIC/GEEC endocytic route supported by Cdc42 and GRAF1.

## 1. Introduction

T cell activation rapidly leads to internalization of T cell receptor (TCR) [1]. Removal of TCR from the cell surface serves to down-modulate its signalling in order to prevent overactivation [2,3] and is correlated with the stimulation outcome during selection in the thymus [4,5]. TCR endocytosis further supports endocytic recycling [6], which is essential to T cell activation [7,8,9,10]. However, the cellular mechanism supporting TCR endocytosis is still debated [1]. Depending on the study, it has been shown that TCR internalization displays features of clathrin-mediated [11,12,13] or clathrin-independent endocytosis [10,12,14,15] (CME and CIE, respectively). Of note, TCR has never been observed in clathrin-coated pits at the plasma membrane and we have shown that inhibition of CME has no effect on TCR internalization. We further have demonstrated that internalized TCR never co-localizes with clathrin-coated vesicles [10].

Beyond the fact that it involves cholesterol-rich membranes [12,14], the CIE pathway mediating TCR internalization has not yet been identified. While CME has been extensively described, CIE is essentially defined as a group of pathways, which do not utilize clathrin to promote membrane invagination and endocytosis [16]. The best characterized of these pathways is the clathrin-independent carriers (CLICs) and GPI-enriched endocytic compartments (GEECs) pathway [16]. Tubular structures, which eventually become endocytic vesicles upon fission from the plasma membrane is a common denominator of the CLIC/GEEC endocytic route. Formation of these tubules and the resulting CLIC internalization of cargoes is powered by actin polymerisation downstream of the small GTPase Cdc42 [16,17,18,19].

Like TCR endocytosis, the Cdc42-mediated CLIC/GEEC pathway relies on cholesterol within the plasma membrane [18]. Crucially, active Cdc42 localizes to the immunological synapse upon T cell activation where it promotes actin polymerization through the Arp2/3 activator WASp [20,21]. However, actin polymerization supporting contractions and morphological changes at the immunological synapse is mediated by another Arp2/3 activator, WAVE, which is downstream of another small GTPase, Rac [22]. Therefore, it is likely that the role of Cdc42 at the immunological synapse is rather to orchestrate exo- and endocytic events than to promote actin-based morphological changes. As a matter of fact, it has been shown that WASp, which is downstream of Cdc42, couples TCR triggering to its internalization [23]. In line with this study, we have shown that TCR endocytosis relies on the activity of Arp2/3 [10].

Hence, we sought to investigate if TCR is taken up through an endocytic route related to the Cdc42-mediated CLIC/GEEC pathway in activated T cells. The CLIC/GEEC endocytic pathway relying on Cdc42 is tightly related to GRAF1, a Bin/amphiphysin/Rvs (BAR) domain-containing protein, which senses and promotes membrane curvature [24]. GRAF1 is recruited to and deactivates Cdc42 by promoting GTP hydrolysis through its GTPase-activating protein (GAP) domain. Cdc42 and GRAF1 generate membrane invaginations, which eventually mature into CLIC endocytic vesicles through a highly dynamic interplay that remains yet to be fully defined [25,26]. Of note, GRAF1 has been shown to contribute to cell polarity, by regulating the delivery and uptake of membrane in the growing and retracting area of the cell [27].

Here we show that GRAF1 delineates tubular invaginations in activated T cells, which are reminiscent of the CLIC endocytic pathway. In line with previous studies, the frequency of these tubules was increased in cells expressing Cdc42-Q61L, a constitutively active form of Cdc42. Our data further revealed that GRAF1-positive tubules resulting from Cdc42-Q61L expression contained TCRζ and were still connected to the plasma membrane, suggesting that TCR is internalized through an endocytic pathway involving Cdc42/GRAF1. We also observed TCRζ in tubules positive for cholera toxin B (CTxB) and CD44, two other markers of the CLIC endocytic pathway. Importantly, the GRAF1-positive tubules did not contain the canonical CME cargo transferrin (Tf). In accordance with these results, expression of Cdc42-Q61L impairs internalization of the TCR-CD3 complex but not of Tf, further indicating that endocytosis of TCR could be mediated by a CLIC pathway relying on Cdc42 and GRAF1.

## 2. Materials and Methods

### 2.1. Expression Constructs

Mammalian expression constructs encoding for GRAF1 and GRAF1-BAR-PH tagged with GFP and Cdc42 and Cdc42-Q61L tagged with mCherry were a gift from R. Lundmark (University of Umeå, Umeå, Sweden). The pRK5-myc-Cdc42-Q61L, pcDNA3-EGFP-Cdc42-wt, pcDNA3-EGFP-Cdc42-T17N, and pcDNA3-EGFP-Cdc42-Q61L (Addgene plasmids #12974, #12975, #12976, and #12986) were gifts from G. Bokoch. TCRζ-mCherry and Lat-mCherry were a gift from K. Gaus (University of New South Wales, Sydney, Australia). PAmCherry and EGFP expression backbones were obtained from Clontech. TCRζ-PAmCherry was made as described in [10]. CD44-GFP was custom-cloned by Genscript Biotech (Piscataway Township, NJ, USA).

### 2.2. Cell Culture and Sample Preparation

Jurkat T cells (#ACC 282) were purchased from the German Collection of Microorganisms and Cell Cultures GmbH (DSMZ, Braunschweig, Germany) and cultured in RPMI 1640 medium (Lonza, Basel, Switzerland) supplemented with 10% (vol/vol) fetal calf serum (FCS) (Gibco Thermo Fisher, Waltham, MA, USA), and 2 mM L-glutamine (Biowest, Nuaillé, France). T cells were transfected 12–24 h prior to the start of experiments using the Neon electroporation kit (Invitrogen Thermo Fisher, Waltham, MA, USA) with 1 µg DNA (single-transfection), 1.2 µg DNA (co-transfections), or 1.5 µg DNA (triple-transfections) per 250,000 cells (2 pulses à 30 ms with 1150 V).

Before imaging the activated cells, Jurkat T cells were incubated for 10 min at 37  °C on 18 mm glass coverslips (Marienfeld, Lauda Königshofen, Germany) that were coated with 10 µg/mL anti-CD3ε (OKT3, 16-0037, eBioscience Thermo Fisher, Waltham, MA, USA) and anti-CD28 (28.2, 16-0289, eBioscience) for 60 min at 37 °C or overnight at 4 °C. To image cells under resting conditions, they were incubated for 20 min on coverslips coated with 0.01% Poly-L-Lysine (P8920, Sigma Aldrich Merck, Darmstadt, Germany) prior to imaging. For live cell imaging without CO_2_, 25 mM HEPES buffer (H0887, Sigma) was added to the samples. Activated cells were imaged within 30 min after their deposition on the coverslips.

For immunofluorescence staining, transfected Jurkat T cells were activated on anti-CD3ε and anti-CD28 coated glass for 20 min and subsequently fixed with fresh, pre-warmed 3.7% paraformaldehyde (PFA) prepared from 16% PFA stock solution (18814, Polysciences, Warrington, PA, USA) for 20 min at 37 °C. After fixation, cells were washed three times with phosphate buffered saline (PBS), permeabilized for 10 min with 100 μg/mL lysolecithin (L5254, Sigma) and washed three times again. Blocking was performed for 1 h in 5% BSA. Subsequently, cells were incubated with 1:75 anti-GRAF1 (NBP1-89732; Novus Biologicals, Littleton, CO, USA), diluted in 0.5% bovine serum albumin (BSA) overnight at 4 °C. After washing three times, cells were incubated for 1 h at room temperature with secondary antibody (1:250 anti-rabbit Alexa Fluor 647, (Life Technologies Thermo Fisher, Waltham, MA, USA), or 1:100 anti-rabbit Alexa Fluor 488, Jackson Immunoresearch, West Grove, PA, USA) in 5% BSA. After washing, samples were mounted on glass slides with fluorescence mounting medium (S302380, Dako Agilent, Santa Clara, CA, USA) and imaged by confocal microscopy.

To record z-stacks of transfected Jurkat T cells and assess surface connection of GRAF1-positive tubules, cells were activated for 20 min on anti-CD3ε and anti-CD28 coated glass at 37 °C. Either cells were directly fixed without further staining or surface exposed plasma membrane was stained with 1:1000 CellMask Deep Red (C10046, Invitrogen Thermo Fisher, Waltham, MA, USA) on ice for 10 min and subsequently fixed with warm 3.7% PFA as described above.

To assess uptake of transferrin in GRAF1-positive tubules, transfected Jurkat T cells were imaged by confocal live cell microscopy in the presence of 20 µg/mL human transferrin Alexa Fluor 546 (Alexa546-Tf, T23364, Invitrogen Thermo Fisher, Waltham, MA, USA) on an activating glass surface starting 10 min after Alexa546-Tf addition.

To label GRAF1-Cdc42 dependent tubules with a cargo, transfected Jurkat T cells were imaged by confocal live cell microscopy in the presence of 1 µg/mL Fluorescein isothiocyanate (FITC)-labelled cholera toxin beta subunit (CTxB-FITC, C1655, Sigma) on an activating glass surface starting 10 min after CTxB-FITC addition.

### 2.3. Flow Cytometry

TCR/Tf internalization assay: Transfected Jurkat T cells were washed once with cold, serum-free Roswell Park Memorial Institute 1640 medium (RPMI). Subsequently, cells were incubated for 30 min at 4 °C in serum-free RPMI with 1.5 µg/mL biotinylated anti-CD3ε (clone OKT3, 13-0037-82, eBioscience or clone SK7, 344820, Biolegend, San Diego, CA, USA) and 1 µg/mL non-biotinylated anti-CD28 (28.2, 16-0289, eBioscience) or with 1.5 µg/mL biotinylated isotype control (13-4724-85, eBioscience) and anti-CD28 (1 µg/mL), or with 30 µg/mL biotinylated transferrin (009-060-050, Jackson Immunoresearch) and non-biotinylated anti-CD3ε (1.5 µg/mL, OKT3) and anti-CD28 (1 µg/mL), to allow antibody/Tf binding but not T cell activation and receptor-internalization. For non-activating conditions, cells were incubated for 30 min at 4 °C in serum-free RPMI with 1.5 µg/mL biotinylated anti-TCRβ (JOVI.1, NBP2-50355B, Novus Biologicals, Littleton, CO, USA). Then, cells were washed three times with cold RPMI, resuspended in either cold RPMI with 10% fetal calf serum (FCS) (T = 0 min) or warm RPMI with FCS (T = 5–30 min) and incubated at 37 °C for the indicated times. Next, cells were washed with cold RPMI. Cells were either directly subjected to Pacific Blue-streptavidin staining (see below), to determine the remaining surface TCR/Tf from initial labelling, or incubated again with biotinylated anti-CD3ε (1.5 µg/mL) or biotinylated Tf (30 µg/mL) in cold, serum-free RPMI for 20 min at 4 °C for determination of the total surface TCR/Tf levels after indicated timepoints. “Total surface” samples were washed three times with cold RPMI and then stained with 4 µg/mL Pacific Blue-streptavidin (S11222, Invitrogen) in PBS for 30 min at 4 °C. After washing with cold PBS, the samples were resuspended in PBS with 2% FCS and measured on a LSR II flow cytometer (Becton-Dickinson, San Jose, CA, USA) equipped with violet (405 nm), blue (488 nm), and red (633 nm) lasers. Samples were kept on ice at all times before measuring. The 0.1 µM TO-PRO-3 (T3605, Invitrogen Thermo Fisher, Waltham, MA, USA) was added directly before measuring for identification of dead cells. Lymphocytes were gated based on scatter signals. Then, live cells were selected in a FSC/TO-PRO-3 dot-plot. Transfected (GFP-expressing) cells were gated and the mean fluorescence intensity of Pacific Blue was determined. Mean fluorescence intensity (MFI) of corresponding isotype controls were subtracted from sample MFI and means of technical duplicates were normalized to the corresponding T = 0 min value or directly plotted.

Calcium-flux: To assess early TCR signalling induced by different antibodies against TCR-complex, calcium-flux assay was performed. Jurkat T cells (10^6^ per sample) were washed once with cold Ca^2+^-flux buffer (145 mM NaCl, 5 mM KCl, 1 mM MgCl_2_, 1 mM CaCl_2_, 1 mM Na_2_HPO_4_, 5 mM glucose, 10 mM HEPES, pH 7.5) and then re-suspended in 1 mL Ca^2+^-flux buffer. The 0.7 µl Fluo-3-AM (4 mM in DMSO, F1242, Invitrogen) was added and incubated for 25 min at room temperature. After washing once with Ca^2+^-flux buffer, the cells were resuspended in 350 µl Ca^2+^-flux buffer and kept at RT for an additional 30 min. Directly before measuring, SYTOX Blue (S34857, Invitrogen Thermo Fisher, Waltham, MA, USA) was added for identification of dead cells. Lymphocytes were gated based on FSC/SSC. Then, live cells were selected in a FSC/SYTOX Blue dot-plot. Baseline of Fluo-3-AM fluorescence was measured for 30 s prior to antibody addition with 1000 events/s. To test T cell activation by different antibody clones against the TCR complex, anti-CD3 ± anti-CD28 (as indicated in the figure legends) were added to the samples to final concentrations between 1 and 10 µg/mL and Fluo-3-AM fluorescence was recorded over time.

### 2.4. Microscopy

Fixed and live cell confocal microscopy were performed on a Leica TCS SP5 II laser-scanning confocal microscope (Leica Microsystems, Wetzlar, Germany), equipped with an argon laser (458 nm, 476 nm, 488 nm, 496 nm, 514 nm), a 543 nm HeNe laser, a 633 nm HeNe laser, and a microscope temperature control system (Life Imaging Services, Basel, Switzerland). GFP constructs and Alexa Fluor 488 stained endogenous GRAF1 were excited using the 488 nm line of the argon laser source, while mCherry-tagged proteins and Alexa546-Tf were excited with the 543 nm laser line. CellMask Deep Red and Alexa Fluor 647 stained endogenous GRAF1 were excited with the 633 nm laser line. Images were acquired with a HCX PL APO CS 63.0×1.40 Oil UV oil-immersion objective (Leica Microsystems, Wetzlar, Germany). For live cell imaging, sequential mode was used with “between lines” settings and bidirectional scanning to enable fast imaging. GFP fluorescence was detected with the HyD detector, mCherry or Alexa Fluor 546 fluorescence was detected with a standard photomultiplier tube. To image fixed samples, sequential mode was used with “between frames” settings and all channels were acquired with the HyD detector. The pinhole was set to 1 Airy Unit.

Live cell total internal reflection fluorescence (TIRF) microscopy used in photoactivation experiments was performed on a Leica DMi8 equipped with an Infinity TIRF module, a 405 nm diode laser, a 488 nm solid state laser, a 561 nm diode pumped solid state laser, and a 638 nm solid state laser. For fast two-channel live cell imaging, the microscope was equipped with an incubation chamber (Pecon, Erbach, Germany) and W-VIEW GEMINI image splitting optics with filters for spectral separation of GFP and mCherry (Hamamatsu, Hamamatsu, Japan). Images were acquired with a HC PL APO 100x/1.47 oil-immersion objective (Leica Microsystems, Wetzlar, Germany) and fluorescence was detected with a DFC9000GTC sCMOS camera (Leica Microsystems). GFP was excited with the 488 nm laser, PAmCherry was photoactivated (PA) with 405 nm laser and excited with the 561 nm laser. TIRF angle was adjusted to 100 nm penetration depth for 488 nm excitation light. PAmCherry in the plasma membrane was photoactivated for 50 frames à 0.2 s with 10% 405 nm laser power. Before, during, and after PA, cells were additionally illuminated with 488 nm and 561 nm lasers simultaneously. Signals from GFP and PAmCherry were separated with the W-VIEW GEMINI image splitting optics and simultaneously recorded on one half of the camera chip.

### 2.5. Image Analysis

For quantification of cells with visible GFP-GRAF1- or GFP-GRAF1-BAR-PH-positive tubules, z-stacks of transfected cells were displayed as maximum intensity projections and visually evaluated. Data analysis was performed blinded.

To quantify the percentage of tubules demarked by GFP-GRAF, cholera toxin beta (CTxB)-FITC or CD44-GFP that contain TCRζ-mCherry, Alexa546-Tf, or Lat-mCherry, confocal time series were visually evaluated. Tubules in the green channel appearing during the recorded time series were visually evaluated if they contained TCRζ-mCherry, Alexa546-Tf, or Lat-mCherry, respectively. Lookup table (LUT) and gamma values of videos were adjusted in order to compensate for variability in brightness due to differing expression levels.

To quantify the number of cells with GFP-GRAF1-positive tubules containing TCRζ-mCherry, cells were transfected to express myc-Cdc42-Q61L, GFP-GRAF1, and TCRζ-mCherry, and imaged on coverslips coated with anti-CD3 and anti-CD28 by confocal microscopy. GFP and mCherry double-positive cells were randomly selected and time series were recorded for 1 min. Videos were evaluated for the presence of GFP-GRAF1-positive tubules and if these tubules also contained TCRζ-mCherry.

### 2.6. Data Analysis

Data from flow cytometry were analysed using FlowJo software v10 (Tree Star, Ashland, OR, USA). Microscopy data were evaluated and prepared for the figures using FIJI software or LAS X v3 (Leica Microsystems, Wetzlar, Germany). All statistical analyses were performed using Prism v7 software (GraphPad, San Diego, CA, USA). Outliers were identified with the ROUT method (Q = 0.1%) and removed from data sets. Unpaired, two-tailed student’s t-test was used to calculate statistical significance. Significance was defined at a 5% level. Not significant (n.s.) *p* > 0.05, * *p* ≤ 0.05, ** *p* ≤ 0.01, *** *p* ≤ 0.001, **** *p* ≤ 0.0001

## 3. Results and Discussion

### 3.1. Cdc42 and GRAF1 Regulate the Formation of Tubular Structures in Activated T Cells

Previous work indicates that GRAF1 is a marker of tubular invaginations leading to CLIC-mediated internalization of cargoes [24,26,27]. In order to investigate if these tubules also form in T cells, we expressed a GFP-tagged GRAF1 in Jurkat T cells and imaged them by confocal microscopy after activation on coverslips coated with activating antibodies against CD3ε and CD28 for 10 min and fixation with 3.7% paraformaldehyde (PFA) at 37 °C. We observed moderate but consistent (8.9 ± 0.9%) formation of GRAF1-positive tubules (Figure 1A,C), indicating that the CLIC pathway marked by GRAF1 occurs in activated T cells. Expressing a mutant of GRAF1 containing the membrane-binding BAR and PH domains but lacking the SH3 and GAP domains (GRAF1-BAR-PH) led to the formation of more tubules compared to WT GFP-GRAF1 (25.5 ± 7.9%), as previously described in Hela cells [26] (Figure 1A,C). Expression of Cdc42-Q61L, a constitutively active form of Cdc42 that is unable to hydrolyse GTP, has been shown to promote the formation of tubular structures demarked by GRAF1, similarly to GRAF1-BAR-PH [25,27]. These tubules are considered to be CLIC invaginations that have failed to mature to endocytic vesicles, due to the inability of GRAF1 to deactivate Cdc42 [25]. Jurkat T cells expressing Cdc42-Q61L showed a marked increase in the frequency of GRAF1-positive tubules compared to cells expressing GRAF1 alone or GRAF1-BAR-PH (40.9 ± 4.7%; Figure 1B,C). Of note, expression of WT Cdc42 had no effect on the formation of GRAF1-positive tubules (Figure 1C). We also could detect endogenous GRAF1-positive tubules in mCherry-Cdc42-Q61L-expressing cells (Figure 1D), using a commercial antibody against GRAF1 that decorated GFP-GRAF1 tubules in test experiments (Appendix A).

Together these data suggest a CLIC pathway relying on Cdc42 and GRAF1 in Jurkat T cells. This is consistent with the idea that active Cdc42 is recruited to the immunological synapse [20,21] in order to regulate endocytic trafficking events. Interestingly, Cdc42 activity is also related to exocytosis through regulation of membrane tension [28,29] or recruitment of secretory vesicles to the plasma membrane [30]. Thus, Cdc42 might globally regulate traffic to and from the plasma membrane at the immunological synapse by coupling endocytic to exocytic events.

### 3.2. GRAF1-Positive Tubules Contain Internalized TCRζ but Not Transferrin

Our results show that Cdc42-Q61L and GRAF1 can be used to generate and visualize tubules related to the Cdc42/GRAF1 endocytic pathway in Jurkat T cells. To investigate if this pathway is involved in the endocytosis of TCR, we expressed a myc-tagged form of Cdc42-Q61L together with GFP-GRAF1 and TCRζ-mCherry in Jurkat T cells. As for Figure 1, these cells were activated on antibody-coated coverslips and imaged in confocal microscopy. We found that 54 ± 33% of GRAF1-positive tubules also contained TCRζ (Figure 2A,D, Appendix A). A global quantification showed that 65.6 ± 15.1% of the cells that expressed Cdc42-Q61L displayed tubules positive for TCRζ (Appendix A). Of note, we also observed TCRζ within GRAF1-positive tubules in cells that did not express Cdc42-Q61L (Appendix A). Furthermore, 33.1 ± 33.5% of GRAF1-positive tubules were positive for TCRζ in cells deposited on non-activating surfaces (Poly-L-Lysine), suggesting that the Cdc42/GRAF1 CLIC pathway is also operating in resting cells (Figure 2D).

By contrast, we identified only 3 ± 7% of GRAF1-positive tubules to be positive for Alexa546-Tf, a canonical CME cargo, in cells expressing Cdc42-Q61L and GFP-GRAF1 and allowed to internalize Tf for 10 min at 37 °C (Figure 2B,D, Appendix A). Accordingly, GRAF1-positive compartments have been reported to be negative for clathrin, Tf, or the Tf receptor [26]. We found that only 11 ± 14% (activated cells) and 0% (resting cells) of the Cdc42-Q61L-induced tubules were positive for the T cell-specific adaptor protein Lat (Figure 2D, Appendix A). The internalization pathway of Lat has not been identified yet, but this result suggests that the pathway mediating the endocytosis of Lat does not depend on Cdc42 and, therefore, is distinct from that of TCR.

To further investigate the connection between TCR endocytosis and the CLIC pathway, we verified if TCR was contained in tubules positive for two previously identified CLIC cargoes, CTxB and CD44 [31,32]. TCRζ was localized in 45 ± 39% of the tubules positive for CTxB in Jurkat T cells expressing Cdc42-Q61L and allowed to internalize CTxB-FITC for 10 min at 37 °C (Figure 2C,D). We could also detect TCRζ-mCherry in 97 ± 9.2% of tubules that were positive for GFP-CD44 (Appendix A). Hence, we observed TCRζ in tubules positive for three different CLIC markers, GRAF1, CTxB, and CD44. This strongly suggests that there is a link between TCR internalization and the CLIC pathway.

These data support the hypothesis that TCR is a CIE cargo internalized through a Cdc42/GRAF1 CLIC pathway, unlike Tf, which enters the cell through CME. To further strengthen this hypothesis, we used a TIRF variation of the microscopy-based internalization assay used previously in order to only visualize endocytosed TCR [10]. We illuminated cells expressing myc-Cdc42-Q61L, GFP-GRAF1, and TCRζ-PAmCherry with 405 nm only within the TIRF evanescent field (< 100 nm) in order to activate PAmCherry essentially at the plasma membrane and not within endosomes (Figure 3A). TCRζ-PAmCherry showed a uniform distribution at the plasma membrane at the end of the photoactivation pulse (Figure 3B, 7 s; Appendix A). At later timepoints, however, intensity of the TCRζ-PAmCherry signal gradually decreased at the plasma membrane, while starting to appear within tubules demarked by GFP-GRAF1 (Figure 3B, from 21 s; Appendix A). These results indicate that TCR contained in GRAF1-positive tubular structures comes from the plasma membrane following endocytosis and not from intracellular stores. This is in agreement with previous work, which showed that CLIC cargoes have been detected in GRAF1-positive tubules shortly after having been incubated with the cells [24].

These results consolidate the hypothesis that TCR is taken up in invaginations of the endocytic route regulated by Cdc42 and GRAF1. We then verified that GRAF1-positive tubules containing TCRζ were still connected to the plasma membrane by using CellMask, a membrane dye designed to be crosslinked by PFA fixation. Cells were activated on antibody-coated glass at 37 °C for 10 min, transferred on ice to prevent any further endocytic process, incubated with CellMask, and fixed with 3.7% PFA after 10 min. This procedure ensured that the membrane making the tubules was decorated by CellMask only if they were still connected to the plasma membrane and open to the extracellular environment. In line with the data obtained using TCRζ-PAmCherry, GRAF1-positive tubules containing TCRζ were also positive for CellMask, suggesting that they resulted from invagination of the plasma membrane related to CLIC endocytic events (Figure 3C).

Endocytic GRAF1-positive tubules have been reported to interact transiently with compartments positive for the membrane-organizing protein flotillins [26]. We have shown previously that TCR is internalized into an endocytic network that is positive for flotillins and that this network promotes the fast recycling of TCR to the immunological synapse through a Rab5-Rab11 endocytic axis [10,33]. Hence, it is possible that TCR internalized through the Cdc42/GRAF1 pathway is eventually transferred to the recycling Rab11-postitive machinery via flotillin-positive compartments.

### 3.3. Expression of Cdc42-Q61L Selectively Impairs Internalization of the TCR-CD3 Complex

Our data show that TCRζ, unlike Tf and Lat, is enriched in GRAF1-positive endocytic tubules in T cells. We next used a flow cytometry-based internalization assay in order to investigate how this translates into endocytosis of the TCR-CD3 complex (Figure 4A, Appendix A). Jurkat T cells were stained with a functional biotinylated antibody against CD3ε and allowed to internalize CD3ε bound to the antibody for the indicated times at 37 °C. The amount of biotinylated anti-CD3ε left at the surface was then detected with a Pacific Blue-labelled streptavidin. At each timepoint, we also detected the total amount of the TCR-CD3 complex present at the cell surface by staining the cells with a combination of the antibody against CD3ε and Pacific Blue-streptavidin (Figure 4A). Endocytosis powered by Cdc42 requires actin polymerization and thus constant switching between hydrolysis of the Cdc42-bound GTP and reloading of GTP [16]. Tubules resulting from the expression of constitutively active Cdc42-Q61L, therefore, cannot mature into vesicular endocytic structures, which stalls the Cdc42/GRAF1-mediated CLIC endocytic route [25]. Accordingly, we found that expressing Cdc42-Q61L in activated Jurkat T cells led to a significant decrease in internalization of the TCR-CD3 complex when compared to cells expressing an empty construct or WT Cdc42, as measured by incorporation of antibodies against CD3ε (clone OKT3 Figure 4B, clone SK7 Appendix A).

By contrast, expression of a dominant negative mutant (Cdc42-T17N) did not influence TCR endocytosis when compared to controls. The lack of effect of Cdc42-T17N on TCR internalization could be explained by a possibly incomplete dominant negative effect in T cells, which have a high expression of endogenous Cdc42. It also has been shown that Cdc42-T17N can be essentially cytosolic and unable to contribute to cellular events taking place at the plasma membrane [18]. More generally, the GRAF1-mediated endocytic pathway is only impaired upon expression of the constitutively active Cdc42-Q61L mutant and not of the dominant negative T17N [25,26]. It appears from these studies that the effect of Cdc42-Q61L on the GRAF1 pathway is less due to the constitutive activity of Cdc42 than to the fact that the Q61L mutation forces GRAF1 to interact with Cdc42 indefinitely. The consequence of this prolonged interaction is that GRAF1 and Cdc42 remain membrane-bound and keep promoting tubulation inside the cell instead of losing membrane association after the initiation of the endocytic invagination. It is possible that other GTPase takes on this short initiation step in cells expressing Cdc42-T17N.

In agreement with the absence of Tf in Cdc42-Q61L-induced tubules (Figure 2B,D), internalization of Tf was not affected by the expression of any form of Cdc42 in activated cells (Figure 4E). Constitutive surface expression of TCR in resting cells expressing Cdc42-Q61L was also increased when compared to the empty vector or WT Cdc42 (Figure 4C), as well as TCR surface levels upon activation, although not significantly (Figure 4D, *p* values = 0.06 and 0.07 at 5 min and 10 min, respectively). Tf surface levels did not show any significant difference between cells expressing the constitutively active form of Cdc42 and the controls in resting or in activated cells (Figure 4F,G).

We further investigated the effect of Cdc42-Q61L expression on TCR endocytosis in resting cells using a non-activating antibody against TCRβ (clone JOVI.1, Appendix A). Resting cells still robustly internalized TCR, although at a more moderate rate than upon activation, especially when compared to the first minutes of activation (Figure 5A,C). The total surface expression of TCR was also decreased in cells expressing Cdc42-Q61L when compared to expression of WT Cdc42 (Figure 5B). Activation-induced endocytosis of TCR was 23% more pronounced than constitutive internalization at 5 min after activation, but only 10.1% at 30 min (Figure 5C). This indicates that TCR internalization tends to return to constitutive values after a boost between 5 and 30 min following TCR triggering. In line with the presence of TCRζ within GRAF1-positive tubules in resting cells (Figure 2D), the expression of Cdc42-Q61L also impaired the internalization of TCR in resting cells (Figure 5A,D). The extent of downregulation induced by the expression of Cdc42-Q61L was similar between resting and activated cells at every timepoint (Figure 5D).

We observed a tendency of Jurkat T cells to upregulate Tf surface expression upon activation, as previously described for the Tf receptor [34]. Altogether, these results indicate that impaired Cdc42/GRAF1-mediated endocytosis in cells expressing Cdc42-Q61L selectively decreases internalization of TCR, and not of Tf, and consequently increases TCR surface expression in resting and activated cells. They are in line with previous work on the role of Cdc42 in endocytosis, which shows that Cdc42 regulates the CLIC/GEEC pathway but has no influence on the internalization of Tf receptor [17]. Interestingly, expression of Cdc42-Q61L decreases the formation of endocytic pits of another clathrin-independent endocytic pathway marked by the BAR-domain containing protein endophilin-A [35]. Furthermore, an in vivo subcellular microscopy study has recently revealed that Cdc42 activity at the plasma membrane may negatively regulate selected endocytic pathways [36]. These reports and our results suggest that too much Cdc42 activity negatively impacts on the internalization pathway downstream of this small GTPase. However, the fact that another study reports the exact opposite (constitutively active Cdc42 having no effect on dextran uptake and expression of a dominant negative Cdc42 decreasing it [17]) rather suggests that Cdc42 activity needs to be fine-tuned in order to mediate endocytic processes. Finally, TCR internalization has been reported to be downstream of another actin-regulating small GTPase, RhoG [15]. In this study, expression of the constitutively active or dominant negative forms of RhoG inhibited TCR endocytosis.

## 4. Conclusions

In this study we showed that the CLIC endocytic pathway regulated by the small GTPase Cdc42 and the BAR-domain containing protein GRAF1 is operative in Jurkat T cells. Our data further indicate that the Cdc42/GRAF1 endocytic route mediates the CIE internalization of TCR through the formation of endocytic tubular structures in activated cells, while playing no role in the uptake of transferrin in the same conditions.

## Figures and Tables

**Figure 1 cells-08-01388-f001:**
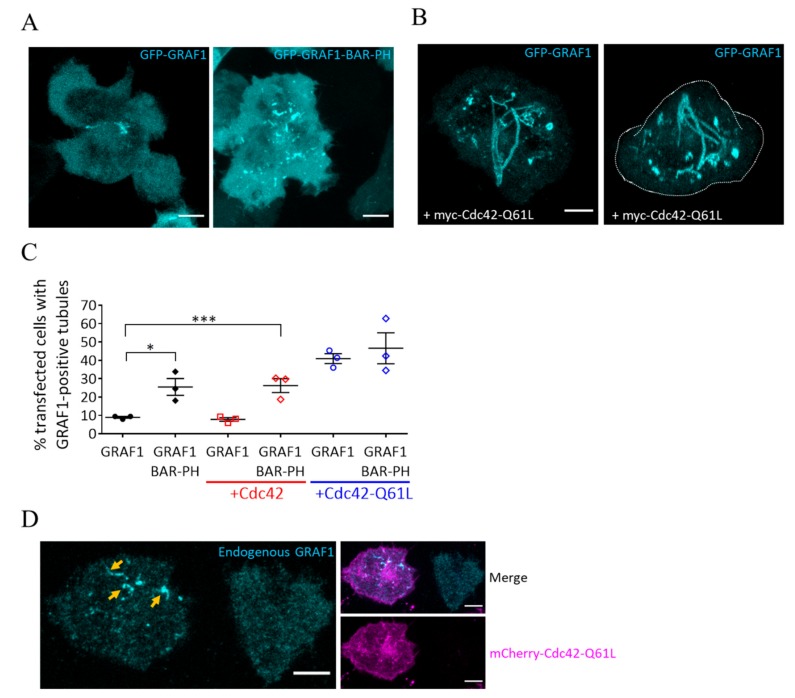
Cdc42 and GRAF1 regulate the formation of tubular structures in activated T cells. (**A**) Jurkat T cells expressing GFP-GRAF1 (left) or GFP-GRAF1-BAR-PH (right) were activated on coverslips coated with functional antibodies against CD3ε and CD28 for 10 min and fixed with paraformaldehyde (PFA). Images are maximum intensity projection of confocal z-stacks (**B**) Left: Maximum intensity projection of a Jurkat T cell expressing GFP-GRAF1 and myc-Cdc42-Q61L and activated as in (**A**). Right: the same cell shown in a 3D reconstruction. (**C**) Blinded quantification of the percentage of cells expressing GFP-GRAF1 and mCherry-tagged Cdc42 variants as indicated and having at least one GRAF1-positive tubule. Each data point represents the mean of an individual experiment, 75–136 cells per experiment. Small horizontal lines indicate mean (±SEM), * *p* < 0.05; *** *p* < 0.001; unpaired, two-tailed student’s t-test. (**D**) Jurkat T cells expressing GFP-Cdc42-Q61L, activated and fixed as in (**A**) and stained with an antibody against endogenous GRAF1. Arrows indicate GRAF1-positive tubules. Images representative of at least three independent experiments. Scale bar: 5 μm.

**Figure 2 cells-08-01388-f002:**
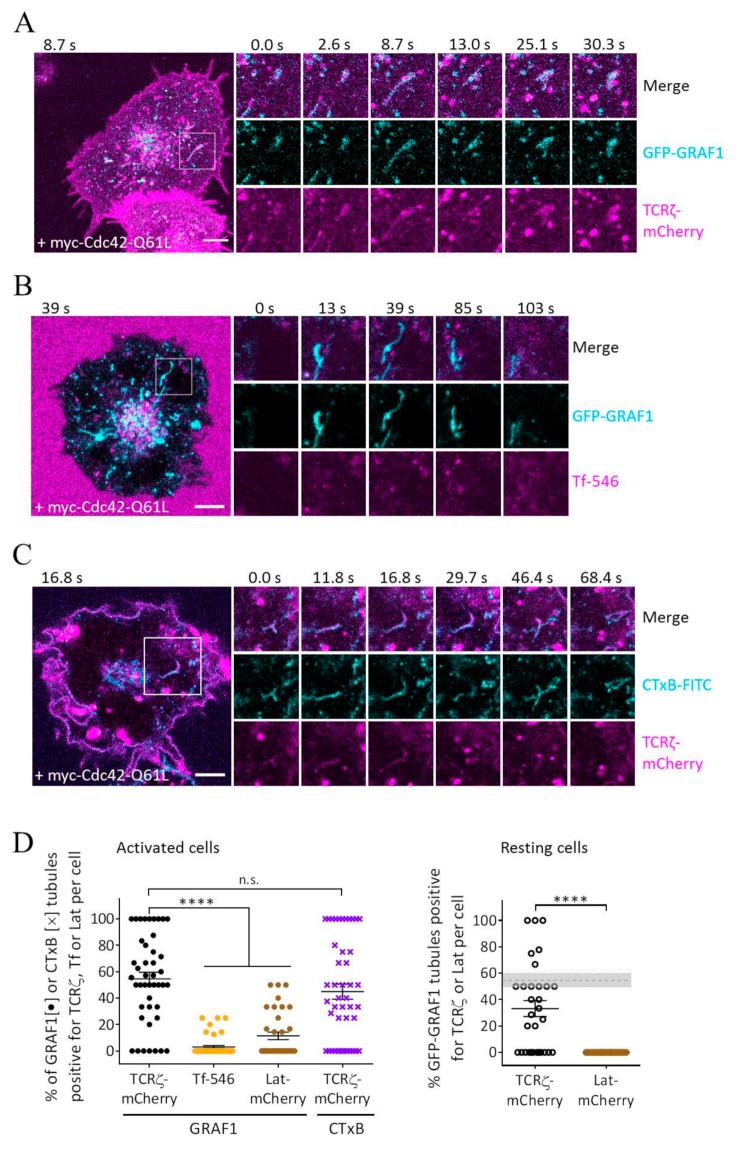
GRAF1- and CTxB-positive tubules contain TCRζ but not transferrin. (**A**) Representative time series of Jurkat T cells expressing myc-Cdc42-Q61L, GFP-GRAF1, and TCRζ-mCherry, and activated on coverslips coated with functional antibodies against CD3ε and CD28 and imaged live at 37˚C. (**B**,**C**) Jurkat T cells expressing myc-Cdc42-Q61L and GFP-GRAF1 (**B**) or TCRζ-mCherry (**C**) were incubated with Alexa546-Tf (**B**) or CTxB-FITC (**C**) for 10 min and imaged live at 37 °C. (**D**) Quantification of the cells as shown in (**A**), (**B**), and (**D**). Percentage of GRAF1- or CTxB-positive tubules that were also positive for TCRζ-mCherry, Alexa546-Tf, or Lat-mCherry in activated (left, anti-CD3ε and anti-CD28) or resting (right, Poly-L-Lysine) cells. On the right, the dotted horizontal line represents the values for TCRζ in activated cells. Data from at least 45 cells in at least 3 independent experiments. Each data point represents a cell. Small horizontal lines indicate mean (±SEM), **** *p  *<  0.0001, unpaired, n.s. not significant, two-tailed student’s t-test. There were 11 outliers removed for Tf, 8 outliers were removed for Lat-mCherry (resting) using the robust regression and outlier removal (ROUT) method (Q = 0.1%). Scale bar: 5 μm.

**Figure 3 cells-08-01388-f003:**
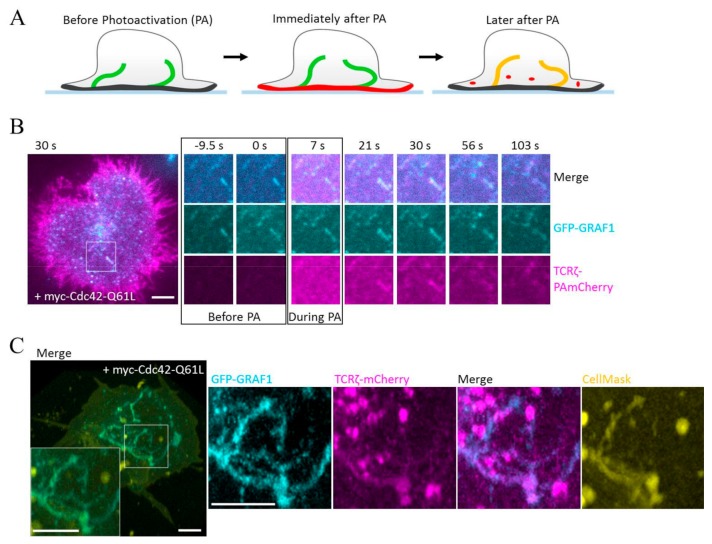
(**A**) Schematic of the PAmCherry-based internalization assay; PAmCherry is photoactivated at the plasma membrane using TIRF illumination for 10 sec. Incorporation of TCRζ-PAmCherry is visualized by the increase in red fluorescence intensity in the GFP-GRAF1 positive tubules. (**B**) Representative time series of the dynamics of the TCRζ-PAmCherry signal before, during, and after photoactivation in Jurkat T cells activated on an antibody-coated coverslip in TIRF illumination. (**C**) Representative image of GFP-GRAF1 and TCRζ-mCherry positive tubules stained with the membrane dye CellMask at 4 °C followed by fixation with PFA at 37 °C. Scale bar: 5 μm.

**Figure 4 cells-08-01388-f004:**
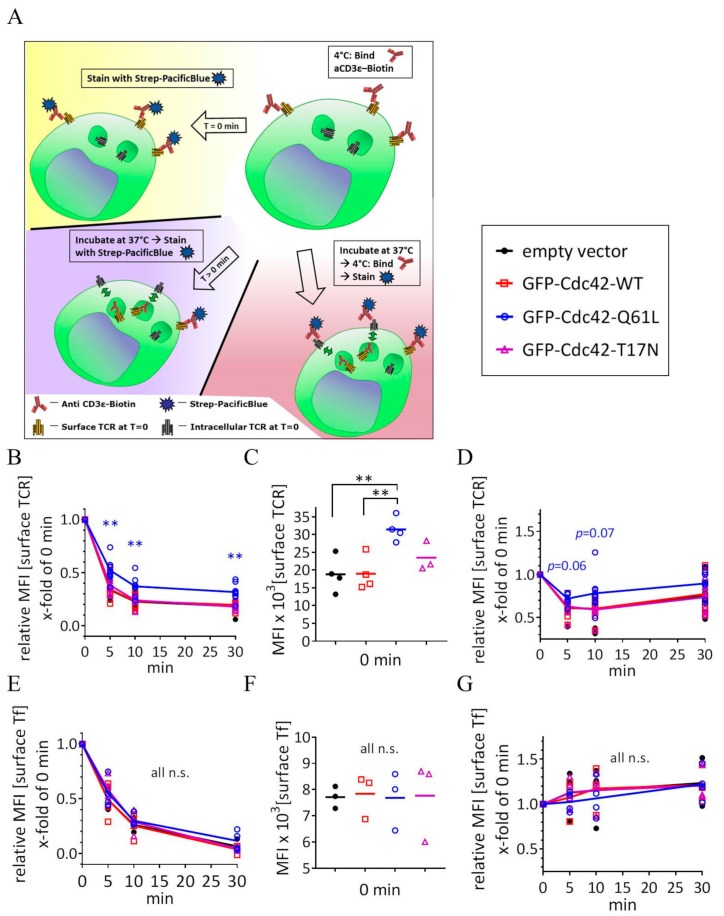
Expression of Cdc42-Q61L selectively impairs internalization of the TCR-CD3 complex. (**A**) Schematic of the flow-cytometry based internalization assay; cells are labelled at 4 °C with a functional biotinylated anti-CD3ε and either (yellow) directly stained with Pacific Blue-streptavidin to measure surface expression of TCR-CD3 complex in resting cells, (blue) activated by incubation at 37 °C and stained with Pacific Blue-streptavidin to detect remaining TCR-CD3 at the cell surface after activation-induced internalization, or (red) activated by incubation at 37 °C, re-labelled with biotinylated anti-CD3ε and stained with Pacific Blue-streptavidin to detect total surface TCR-CD3 in activated cells. (**B**) Remaining TCR-CD3 at the cell surface detected by an antibody against CD3ε (clone OKT3) after activation-induced internalization in cells expressing an empty vector, GFP-WT-Cdc42, GFP-Cdc42-Q61L, or GFP-Cdc42-T17N. (**C**) Surface expression of TCR-CD3 complex in cells expressing the same constructs as in (**B**). (**D**) Total surface TCR-CD3 in activated cells transfected as in (**B**). (**E**) Internalization of biotinylated Tf detected with Pacific Blue-streptavidin after incubation at 37 °C as described for anti-CD3ε in (**A**), in cells activated by soluble anti-CD3ε and expressing the same constructs as in (**B**). (**F**) Surface expression of Tf in cells expressing the same constructs as in (**B**). (**G**) Total Tf at the cell surface detected with biotinylated Tf and Pacific Blue-streptavidin in activated cells as in (**F**). Each data point represents the mean of an individual experiment. Small horizontal lines indicate mean (±SEM), ** *p*  <  0.01, n.s. not significant, unpaired, two-tailed student’s t-test.

**Figure 5 cells-08-01388-f005:**
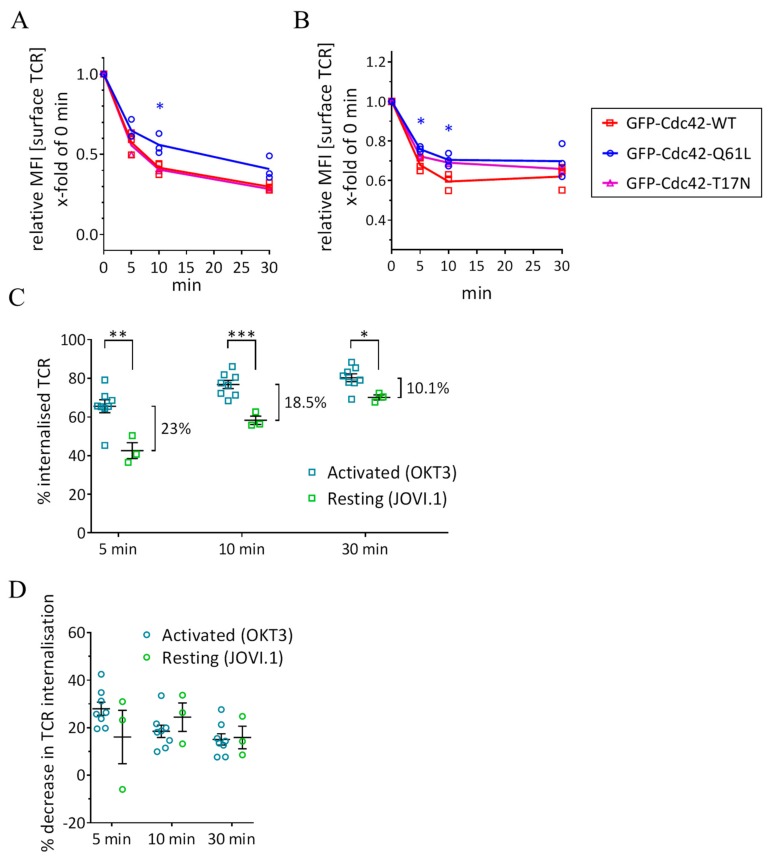
Resting T cells constitutive internalization of the TCR-CD3 complex is impaired by Cdc42-Q61L expression. (**A**) Remaining surface TCR-CD3 detected by an antibody against TCRβ at the indicated timepoints in cells expressing GFP-Cdc42-WT, GFP-Cdc42-Q61L, or GFP-Cdc42-T17N. (**B**) Total surface TCR-CD3 in resting cells expressing the same constructs as in (**A**) over time. (**C**) Percentage of TCR internalized from cell surface in respect to the amount detected at t = 0 in activated cells (anti CD3ε, data from Figure 4B) or resting cells (TCRβ) at the indicated time points. (**D**) Extent of impairment in TCR internalization upon expression of GFP-Cdc42-Q61L (in %) in respect to expression of GFP-Cdc42-WT in activated or resting cells. Each data point represents the mean of an individual experiment. Small horizontal lines indicate mean (±SEM), * *p*  <  0.05, ** *p*  <  0.01, *** *p*  <  0.001, unpaired, two-tailed student’s t-test.

## Data Availability

All data is deposited on Zenodo and is publicly available, doi: 10.5281/zenodo.3524892.

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
