# Peer review of "Cdc42 Couples T Cell Receptor Endocytosis to GRAF1-Mediated Tubular Invaginations of the Plasma Membrane"

_cells, 2019, doi:10.3390/cells8111388_

Round 1
Reviewer 1 Report
This is a very interesting paper that investigates the mechanism of entry of TCR. The authors present data on the involvement in this process of the CLIC/GEEC pathway and of CDC42 and GRAF1. The data are nicely presented and convincing as the authors, using different tecniques, were able to demonstrate that the CLIC/GEEC pathway is present in Jurkat cells and is responsible for internalization of TCR. Modulating expression or using mutant proteins they demonstrated that CDC42 and the BAR-domain containing protein GRAF1 regulate TCR internalization while transferrin endocytosis is not affected. The paper certaintly deserves publication in Cells.
Mayor point
The lack of effect of the expression of the dominant negative CDC42 mutant is quite surprising. The explanation of the authors is not convincing as, independently of its cytosolic localization, the mutant has been demonstrated to be a dominant negative mutant (as it is the case for many other small GTPase dominant negative mutants that are mainly cytosolic but are dominant negative probably because they sequester effectors in the cytoplasm and/or are not able to stimulate them to act). I understand that it is not easy to design experiments to establish why this mutant in this pathway does not exert its inhibitory function but the authors should at least try to discuss more deeply the issue. One possible explanation could be the presence of other small GTPases that make up for the function of Cdc42.
Minor points
-In the conclusions the authors talk about the CLIC/GRAF1 pathway: shouldn't it be CLIC/GEEC?
-There should be a separation between Figure 1 legend and the main text (lines 269-270)
-"doted" should be "dotted" (line 318)
Author Response
Cdc42 couples T cell receptor endocytosis to GRAF1-mediated tubular invaginations of the plasma membrane
Point-by-point response to referees’ comments
Figure and line numbering refer to the revised version of the manuscript. Changes in the manuscript are highlighted in yellow.
Reviewer 1
The lack of effect of the expression of the dominant negative CDC42 mutant is quite surprising. The explanation of the authors is not convincing as, independently of its cytosolic localization, the mutant has been demonstrated to be a dominant negative mutant (as it is the case for many other small GTPase dominant negative mutants that are mainly cytosolic but are dominant negative probably because they sequester effectors in the cytoplasm and/or are not able to stimulate them to act). I understand that it is not easy to design experiments to establish why this mutant in this pathway does not exert its inhibitory function but the authors should at least try to discuss more deeply the issue. One possible explanation could be the presence of other small GTPases that make up for the function of Cdc42..
We agree with the reviewer, the lack of effect of this mutant is surprising and we have not been able to design experiments to address this specific question.
However, we want to point out that none of the work previously done on the GRAF1-mediated endocytic route has reported an effect of Cdc42-T17N on this pathway. Only Cdc42-Q61L has been shown to stall the formation of GRAF1-positive endocytic vesicles. Francis et al. did not observe any consequence of Cdc42-T17N expression on the formation of tubules or dextran uptake [1]. Similarly, Lundark et al report that only Cdc42-Q61L has an effect on tubule formation [2]. Moreover, we could find only two examples of a negative effect of Cdc42-T17N on the CLIC/GEEC pathway in the literature [3,4]. Of note, reference [4] reports a decrease of dextran endocytosis upon Cdc42-T17N expression, but no effect of Cdc42-Q61L, illustrating that the effect of these mutants can be cell or cargo specific.
We also want to mention that we used Cdc42-Q61L primarily to stall GRAF1-positive endocytic invaginations, in order to investigate if TCR endocytosis would be related to this pathway. Indeed, the contribution of Cdc42 to this pathway seems very complex and remains to be fully elucidated, but this was not the purpose of the submitted manuscript.
It seems that the effect of Cdc42-Q61L on the GRAF1 pathway is less due to the constitutive activity of Cdc42 than to the fact that the Q61L mutation forces GRAF1 to interact with Cdc42 indefinitely [1]. As stated in a detailed study, “GRAF1-mediated inactivation of Cdc42 is not vital for the formation and budding of endocytic carriers, but for their further maturation” [1]. In short, because of the Q61L mutation, both GRAF1 and Cdc42 remain bound to the membrane and keep promoting its tubulation inside the cell, when this step is normally very transient and is associated with the detachment of both GRAF1 and Cdc42 from the membrane after initiation of the invagination.
As suggested by the reviewer, it is very likely than another GTPase takes on this short step in cells expressing Cdc42T17N, which would explain why it has never been reported to have any effect on the GRAF1-mediated endocytic route. Following the reviewer’s suggestion, we have discussed this point in greater length in the revised version of our manuscript (lines 402-410)
Minor points
In the conclusions the authors talk about the CLIC/GRAF1 pathway: shouldn't it be CLIC/GEEC?
We agree with the reviewer, this sentence was confusing. We wanted to refer to GRAF1 contribution only in the context of the endocytic part of this pathway, the CLathrin-Independent Carrier (CLIC), and not to the intracellular component, the GPI-Enriched Endocytic Compartments (GEEC). We have changed the text to avoid any confusion. It now reads: ”…the CLIC endocytic pathway regulated by the small GTPase Cdc42 and the BAR-domain containing protein GRAF1...” (line 489)
There should be a separation between Figure 1 legend and the main text (lines 269-270)
The page layout is not final, and has changed after we submitted the manuscript. Nevertheless, we have added a space to facilitate the reading of the revised manuscript.
"doted" should be "dotted" (line 318).
We thank reviewer for spotting this typo, which we have now corrected (line 327).
References
Francis, M.K.; Holst, M.R.; Vidal-Quadras, M.; Henriksson, S.; Santarella-Mellwig, R.; Sandblad, L.; Lundmark, R. Endocytic membrane turnover at the leading edge is driven by a transient interaction between Cdc42 and GRAF1. J. Cell Sci. 2015, 128, 4183–95. Lundmark, R.; Doherty, G.J.; Howes, M.T.; Cortese, K.; Vallis, Y.; Parton, R.G.; McMahon, H.T. The GTPase-activating protein GRAF1 regulates the CLIC/GEEC endocytic pathway. Curr. Biol. 2008, 18, 1802–8. Gauthier, N.C.; Monzo, P.; Kaddai, V.; Doye, A.; Ricci, V.; Boquet, P. Helicobacter pylori VacA Cytotoxin: A Probe for a Clathrin-independent and Cdc42-dependent Pinocytic Pathway Routed to Late Endosomes. Mol. Biol. Cell 2005, 16, 4852–4866. Sabharanjak, S.; Sharma, P.; Parton, R.G.; Mayor, S. GPI-anchored proteins are delivered to recycling endosomes via a distinct cdc42-regulated, clathrin-independent pinocytic pathway. Dev. Cell 2002, 2, 411–23.

Reviewer 2 Report
Authors found that internalised TCR accumulates in tubules in T cells but the canonical clathrin cargo transferrin or the adaptor protein Lat does not. These tubules are shaped by the small GTPase Cdc42 and GRAF1. Preventing GRAF1-positive tubules to mature into endocytic vesicles by expressing a constitutively active Cdc42 impairs the endocytosis of TCR, while having no consequence on the uptake of transferrin. Authors reveal a link between TCR internalisation and the CLIC/GEEC endocytic route supported by Cdc42 and GRAF1.
These findings are novel and all experiments seem to be sound and obtained results are interesting. Reading this manuscript was fun for me.
This reviewer agree with most part of this manuscript but still have some concerns.
Several concerns.
Spelling.
I think “internalization” would be suitable instead of “internalization” used in this manuscript. Am I wrong?
Yellow marking.
What does mean yellow marking in the text?
Line 241
“Expressing a mutant of GRAF1 containing the membrane binding BAR and PH domains but lacking the SH3 and GAP domains”
Is this mutant of GRAF1 is a constitutively active form or dominant negative form or non-functional form? Please explain.
Line 246
What does mean “GRAF1-BAR-PH”? There are no explanation.
I assume that GRAF1 containing the membrane binding BAR and PH domains but lacking the SH3 and GAP domains (GRAF1-BAR-PH).
In Figure 1 D
What do mean yellow arrow and endogenous GRAF1-488?
GRFA1 which contains aa1-488, or endogenous GRAF1 was detected using Alexa 488?
Please explain.
Line 298
They found that only 3% ± 7% of GRAF1-positive tubules to be positive for Alexa546-Tf and only 11% ± 14% (activated cells) and 0% (resting cells) of the GRAF1-tubules were positive for the T cell-specific adaptor protein Lat in cells expressing Cdc42-Q61L and GFP-GRAF1.
How do authors conclude that “this result suggests that Lat endocytosis is not mediated by Cdc42 and therefore is distinct from that of TCR”? I cannot well catch their logic how authors conclude “not mediated by Cdc42”.
Lien 388
They said that Cdc42-T17N is a dominant negative mutant, however, did not influence TCR 388 endocytosis when compared to controls.
I think it is necessary to check loss of CdC42. I wonder what happens if authors use knock-down instead of Dominant Negative form?
In Figure 5. A
Left panel is probably remaining TCR-CD3 detected by an antibody against TCR? 454 at the cell surface at the indicated time points.
How about right panel? Is total surface TCR-CD3 assumed from Fig.4 or not? Please explain and label the panel.
In supplemental Figure S1.
What does mean GRAF1-647?
Minor points.
Line 76
“CD44, two should be “CD44, two”.
Line 269 and 270 and line 420 and 421
Between these two lines, please make one space.
Author Response
Cdc42 couples T cell receptor endocytosis to GRAF1-mediated tubular invaginations of the plasma membrane
Point-by-point response to referees’ comments
Figure and line numbering refer to the revised version of the manuscript. Changes in the manuscript are highlighted in yellow.
Reviewer 2
This reviewer agree with most part of this manuscript but still have some concerns. Several concerns.
Spelling. I think “internalization” would be suitable instead of “internalization” used in this manuscript. Am I wrong?
“Internalisation”, like “organisation”, or “signalling” is UK English spelling, which we prefer to use over the US spelling (“Internalization”, “organization”, or “signaling”)
Line 241
“Expressing a mutant of GRAF1 containing the membrane binding BAR and PH domains but lacking the SH3 and GAP domains”. Is this mutant of GRAF1 is a constitutively active form or dominant negative form or non-functional form? Please explain.
This mutant, as described in the text, lacks the SH3 and GAP domains and consists essentially of the membrane binding BAR and PH domains [1]. It has been shown to promote the formation of tubules [1], and we used it in this context. It has been hypothesised that this mutant could potentially act as a dominant negative GRAF1, but never formally demonstrated, this is why we do not mention it as such.
Line 246
What does mean “GRAF1-BAR-PH”? There are no explanation. I assume that GRAF1 containing the membrane binding BAR and PH domains but lacking the SH3 and GAP domains (GRAF1-BAR-PH).
The reviewer is twice correct: 1) we mean the mutant of GRAF1 lacking the SH3 and GAP domains described above. And 2), we have indeed forgotten to define this abbreviation. We have revised the text accordingly (line 247).
In Figure 1 D What do mean yellow arrow and endogenous GRAF1-488?. GRFA1 which contains aa1-488, or endogenous GRAF1 was detected using Alexa 488?
We indeed mean endogenous GRAF1 detected with a primary antibody against GRAF1 and a secondary antibody labelled with Alexa-488. We agree with the reviewer, this labelling was not very clear. We have removed the reference to 488 to avoid any confusion. The arrows indicate tubules and we had indeed forgotten to describe them in the figure legend. This has now been corrected (line 274).
Line 298. They found that only 3% ± 7% of GRAF1-positive tubules to be positive for Alexa546-Tf and only 11% ± 14% (activated cells) and 0% (resting cells) of the GRAF1-tubules were positive for the T cell-specific adaptor protein Lat in cells expressing Cdc42-Q61L and GFP-GRAF1.
How do authors conclude that “this result suggests that Lat endocytosis is not mediated by Cdc42 and therefore is distinct from that of TCR”? I cannot well catch their logic how authors conclude “not mediated by Cdc42”.
The hypothesis that we want to express in the submitted manuscript is that Cdc42 and GRAF1 contribute to the endocytosis of TCR. Accordingly, we show TCR is in tubules induced by Cdc42-Q61L. We also show that transferrin, whose internalisation pathway is distinct from TCR and not mediated by Cdc42, is not in these tubules. Finally, we show that Lat, whose internalisation pathway remains unknown, is not in the Cdc42-Q61L-induced tubules, like transferrin. For us, this suggests that Lat, like transferrin, does not use a Cdc42-mediated endocytic route, unlike TCR. We have amended the main text to make this point easier to understand (lines 304-308)
Lien 388. They said that Cdc42-T17N is a dominant negative mutant, however, did not influence TCR 388 endocytosis when compared to controls. I think it is necessary to check loss of CdC42. I wonder what happens if authors use knock-down instead of Dominant Negative form?
We used siRNAs against Cdc42 to silence its expression in Jurkat T cells. Unfortunately, we could never get a higher downregulation than 62% (see Figure R1 in this document). T cells express very high levels of Cdc42 and consequently the 38% remaining still represented a high amount of available Cdc42. Accordingly, we could not measure any significant difference in TCR uptake between control cells and cells in which Cdc42 had been silenced (see Figure R1).
We have nevertheless discussed this point in greater length in the revised version of the manuscript (lines 402-410).
[Here should be a figure, but unfortunately the MDPI system does not allow to display it. Please use the uploaded PDF file]
Figure R1. A 40% downregulation of Cdc42 has no impact on TCR uptake in activated Jurkat T cells. A. Representative image and quantification of western blots (n = 2) of cell lysates made from Jurkat T cells electroporated with siRNA against GFP (control) or against Cdc42 after 72h or 96h. B. Flow cytometry-based uptake measurements of the Jurkat T cells treated as in A.
In Figure 5. A. Left panel is probably remaining TCR-CD3 detected by an antibody against TCR? 454 at the cell surface at the indicated time points. How about right panel? Is total surface TCR-CD3 assumed from Fig.4 or not? Please explain and label the panel.
We thank the reviewer for having spotted that the figure legend of Figure 5 was incomplete. We have re-labelled the figure panels, completed the figure legend and added a line in the main text to refer better to this panel (lines 446-447).
In supplemental Figure S1. What does mean GRAF1-647?
Same as previously, it means endogenous GRAF1 detected with a primary antibody against GRAF1 and a secondary antibody labelled with Alexa-647. We have removed the reference to 647 to avoid any confusion
Minor points. Line 76. “CD44, two should be “CD44, two”.
The reviewer is correct; there was a mistake with the font size, which we have now corrected (line 76).
Line 269 and 270 and line 420 and 421. Between these two lines, please make one space.
The page layout is not final, and has changed after we submitted the manuscript. Nevertheless, we have added these spaces to facilitate the reading of the revised manuscript.
Reference
Lundmark, R.; Doherty, G.J.; Howes, M.T.; Cortese, K.; Vallis, Y.; Parton, R.G.; McMahon, H.T. The GTPase-activating protein GRAF1 regulates the CLIC/GEEC endocytic pathway. Curr. Biol. 2008, 18, 1802–8.
This manuscript is a resubmission of an earlier submission. The following is a list of the peer review reports and author responses from that submission.
Round 1
Reviewer 1 Report
The manuscript presented by Rossatti et al. propose that TCR endocytosis occurs through a GRAF1-CdC42 regulated pathway in line to the many CLIC-GEEC pathways previously proposed.
While potentially interesting the paper fail to convince the reviewer and the reviewer encourage the authors to reconsider the observations.
This as nothing to do with the fact that TCR enters by a non clathrin endocytic pathway, point already made in previous study from the senior author (PMID: 29686427), but rather to the demonstration of entry through a GRAF-CdC42 mediated pathway.
From the reviewer opinion the data presented only weakly support a potential link with this (existing in jurkat cells?) pathway.
The major problem lies in the fact that the reviewer is not convince of even the existence of this pathway from the data presented in figure 1 and 2 were cells present few tubules that are potentially decorated by GRAF1 and reinforced by dominant positive cdc42.
The reviewer also z projected the video 1 in max average (imageJ) to unveil the tracks followed by GRAF or the TCR (see attached file), there is almost no colocalization from the large majority of the tracks between the 2 molecules.
The dynamic presented in this paper is also not consistent with the previous report (PMID: 29686427) where, upon photoactivation, TCR enter much faster.
I think the author have potentially many movie and nice tools in hands (photoactivation is a fantatstic tool) to try to decorticate better the pathway of endocytosis of TCR and should rework on that, trusting their data and tools, rather than trying to fit endocytosis of TCR with a somehow confusing and unclear potential pathway regulated by GRAF1 and proposed by other people.
Also to note, the coverslip-coated assay is very hard to correlate with the suspended assay, as cells are in completely different state and it will be good to try to image jurkat in a suspended-like configuration to clearly make the link between the FACS data and imaging.

Reviewer 2 Report
In this study, Rossatti et al. investigate the mechanisms regulating the endocytosis of the T cell receptor (TCR). The results suggest that TCR is internalized into cells through the CLIC/GEEC pathway that is regulated by the small GTPAse Cdc42. The manuscript is sometimes difficult to follow. It would help readers whether authors structure their text with shorter paragraphs. The statement and conclusion are supported by a rather low amount of work and would deserve to be further documented with complementary approaches. It is an interesting story, but under the current form, this work rather appears as under progress. For instance, showing direct colocalization between CLIC and TCR would be an asset, the same with testing potential direct colocalization between transferrin and TCR. Overall, a systematic comparison between resting and activated T cells is missing and would reinforce the idea that this uptake mechanism requires cell activation. Authors mentioned several times the possibility that TCR is recycled to the plasma membrane through a Cdc42-dependent process but did not test the role of the two classical Rabs 4 and 11, which are well characterized for their function in the early and late recycling pathways from plasma membrane. Authors highlighted the weak efficiency of dominant negative mutant of Cdc42. Could they confirm their results in cells silenced for Cdc42? If not possible to silence the gene, authors should explain why. There are a couple of minor points that should be addressed (see below).
Minor points:
1. Line 39, it is written “clathrin pits”, however my guess is that authors mean “clathrin-coated vesicles”.
2. Lines 39-40, it is written that TCR internalization is incompatible with CME kinetic. Could author explicitly explain why.
3. Often the style for the references to figures is not homogenous, e.g. line 144 “FigS3 and line 246 “Supplemental Figure S2B, C”.
4. Lines 260 – 268, TIRF results are difficult to understand. If PAmCherry is really internalized, it should disappear from the plasma membrane and tubules should not be visible in the same plan. Could authors clarify?
5. Is CellMask really membrane impermeant?
6. Line 298, “defined by flotillins” should be better explained.
7. Lines 317-320, authors mention that expression of Cdc42-Q61L results in a decrease in internalization of TCR-CD3 into activated T cells. One could expect that uptake is not impaired but instead proteins accumulate in the tubules. So close from the plasma membrane, it is difficult to distinguish between what remains at the surface or is taken up but blocked within nascent tubules underneath the plasma membrane. Could authors comment?
8. Line 363, it is not clear to me about which results authors are talking to support that a too high activity negatively impacts the internalization downstream of Cdc42.
9. Line 366, if I remember well, dextran is a cargo of the macropinocytotic pathway.
